# The Potential of Flavonoids for the Treatment of Neurodegenerative Diseases

**DOI:** 10.3390/ijms20123056

**Published:** 2019-06-22

**Authors:** Pamela Maher

**Affiliations:** Salk Institute for Biological Studies, La Jolla, CA 92037, USA; pmaher@salk.edu; Tel.: +1-858-453-4100

**Keywords:** oxidative stress, cognitive dysfunction, inflammation, cell death, synapse loss, protein aggregation, neurodegenerative disease

## Abstract

Neurodegenerative diseases, including Alzheimer’s disease (AD), Parkinson’s disease (PD), Huntington’s disease (HD), and amyotrophic lateral sclerosis (ALS), currently affect more than 6 million people in the United States. Unfortunately, there are no treatments that slow or prevent disease development and progression. Regardless of the underlying cause of the disorder, age is the strongest risk factor for developing these maladies, suggesting that changes that occur in the aging brain put it at increased risk for neurodegenerative disease development. Moreover, since there are a number of different changes that occur in the aging brain, it is unlikely that targeting a single change is going to be effective for disease treatment. Thus, compounds that have multiple biological activities that can impact the various age-associated changes in the brain that contribute to neurodegenerative disease development and progression are needed. The plant-derived flavonoids have a wide range of activities that could make them particularly effective for blocking the age-associated toxicity pathways associated with neurodegenerative diseases. In this review, the evidence for beneficial effects of multiple flavonoids in models of AD, PD, HD, and ALS is presented and common mechanisms of action are identified. Overall, the preclinical data strongly support further investigation of specific flavonoids for the treatment of neurodegenerative diseases.

## 1. Introduction-What Is Neurodegeneration?

Before reviewing the potential beneficial effects of natural products, and in the case of this review, specifically flavonoids, on neurodegeneration, it is essential that a definition of neurodegeneration be established. Over 15 years ago, Przedborski et al. [1] published a comprehensive discussion of this topic that is still highly relevant today. They defined neurodegeneration generally as “any pathological condition primarily affecting neurons”. More specifically, they characterized neurodegenerative diseases as a large, heterogeneous group of neurological disorders that affect distinct subsets of neurons in specific anatomical locations. They also noted that a number of disorders that are either not primary neuronal diseases or where neurons die of a known cause, such as hypoxia or poison, are not neurodegenerative diseases. While hundreds of neurodegenerative disorders are known, most of the attention has focused on four: Alzheimer’s disease (AD), Parkinson’s disease (PD), Huntington’s disease (HD), and amyotrophic lateral sclerosis (ALS), although others, such as frontotemporal dementia (FTD), are as common, if not more so, than either HD or ALS [2]. In the United States, there are currently 5.8 million people with AD [3], over 700,000 with PD [4], ~30,000 with HD [5], ~16,000 with ALS [6], and 50,000–60,000 with FTD [2]. For AD, PD, and ALS, there are both genetic and sporadic forms of the disease, with the vast majority of the cases of all three being sporadic. In contrast, FTD has a stronger genetic component [2] and almost all cases of HD are dominantly inherited [5]. Regardless, for all of these diseases and irrespective of the cause, the strongest risk factor for developing any of them is increasing age. This suggests that changes that occur in the aging brain put it at increased risk for the development of a neurodegenerative disease and that the identification of those changes could provide a means to develop therapeutics that can at least slow, if not prevent, disease development and/or progression.

### 1.1. Aging and Age-Associated Changes in the Brain

Among the pathophysiological changes that occur in the aging brain, those that have been identified as potentially contributing to neurodegeneration include increases in oxidative stress, alterations in energy metabolism, loss of neurotrophic support, alterations in protein processing leading to the accumulation of protein aggregates, dysfunction of the neurovascular system, and immune system activation [7,8]. Given this multiplicity of changes, it is unlikely that targeting a single change will prove effective at preventing nerve cell damage and death. In addition, there is a strong possibility that the relative contributions of each of these changes will vary among individuals. Importantly, these changes interact with lifestyle, environmental, and genetic risk factors with varying degrees of penetrance. For example, although AD is defined in terms of plaque and tangle pathology, it is most frequently associated with other detrimental events, such as microvascular damage and inflammation [9]. Thus, it is likely it will be necessary to use combinations of drugs directed against different targets in order to effectively prevent these age-related changes to the brain. However, this approach is subject to a number of potential problems, including pharmacokinetic and bioavailability challenges, which in central nervous system (CNS) diseases are exacerbated by the difficulty of getting multiple compounds across the blood brain barrier and the potential for adverse drug–drug interactions. A better approach is to identify small molecules that have multiple biological activities that can impact the multiplicity of age-associated pathophysiological changes to the brain that contribute to neurodegenerative disease development and progression [10].

### 1.2. Approaches to Drug Discovery for Neurodegenerative Diseases

Since the 1990s, the combination of molecular and structural biology, combinatorial chemistry, and high throughput screening has dominated the drug discovery process [11]. This approach provides a rapid process for the discovery of drug candidates with high selectivity and high affinity for a specific molecular target. However, it has not produced the successes that were initially expected, especially with respect to complicated problems such as neurodegenerative diseases. Prior to the development of this target-based drug discovery approach, new drugs were discovered by evaluating chemicals against observable characteristics or phenotypes, in biological systems such as cells or animals. While this approach has fallen out of favor with the pharmaceutical industry, surprisingly a recent study showed that it still continues to be more successful than target-based approaches for the identification of first-in-class small molecule drugs [12]. It has been argued that this is because target-based discovery is based on a priori assumptions that do not take into account the complexities of biological systems or diseases [7,12].

The ideal phenotypic drug screening paradigm would employ the ultimate end user—humans—and this is how most of the natural product-based, first-in-class drugs were originally discovered. However, this is no longer an ethically viable approach. Laboratory animals, primarily disease models in mice, are currently used for preclinical testing but using them for the initial screening of drug candidates is impractical due to cost and time constraints, as well as the drive to reduce animal use in research. A reasonable alternative is to create cell-based assays that define molecular toxicity pathways relevant to age-associated neurodegeneration and select drug candidates that work in multiple assays, not just one [7]. In this way, the screening paradigms have disease relevance, reproducibility, and reasonable throughput. Many arguments can be made against the relevance of any single cellular screening assay based on the cell type or the nature of the toxic insult. Thus, to account for individual weaknesses, phenotypic screening paradigms for neurodegenerative diseases should combine multiple assays that address the different toxicities associated with the aging brain. This enables the identification of potent, disease-modifying compounds for preclinical testing in animal models of neurodegenerative diseases. In general, for screening for drug candidates against neurodegenerative diseases, these assays will utilize primary neurons, neuron-like cell lines, or microglial cell lines that are subjected to a toxic insult that has been observed to occur in the aging brain. However, the critical question still remains of what exactly should be screened.

### 1.3. What to Start with

One excellent source for multi-target compounds is the original pharmacopeia—plants. The earliest records describing the use of plants for medicinal purposes date back to 2600–2900 BC [13]. Still today, ~25% of all prescribed drugs are thought to be derived from plants [14]. Plants synthesize a huge array of compounds called secondary metabolites that are not required for plant growth. These compounds are derived from a limited number of basic chemical scaffolds, which are modified by specific types of substitutions. It has been suggested [14] that these compounds, as well as receptors, enzymes, and regulatory proteins, originated from a relatively small number of parental molecules, which may have co-evolved to interact with one another. Although their biological functions and structures have since diverged, structural features shared from their common past may be the reason that they interact with medically relevant targets.

### 1.4. Why Focus on Flavonoids

Among the huge number of plant-derived secondary metabolites, several epidemiological studies have specifically highlighted the potential beneficial role of flavonoids for the prevention of neurodegenerative diseases. The over 5000 different flavonoids can be divided into six groups (flavones, flavonols, flavanones, flavanols, anthocyanidins, and isoflavones) based on the degree of oxidation of the central C ring, the hydroxylation pattern of the rings, and the substitution at the 3 position (Figure 1). Within each group, diversity is generated by the arrangement of the hydroxyl groups combined with glycosylation or alkylation [15].

A retrospective study that looked at flavonoid intake versus disability adjusted life years (a measure of the burden that a disease has on those affected in a population) due to dementia in 23 developed countries found that total combined flavonoid intake was significantly and negatively correlated with dementia [16]. Among the flavonoid groups, only flavonol consumption showed a significant, negative correlation with dementia. Consistent with these results, a prospective cohort study [17] found that the risk ratio for dementia (risk of dementia in the high flavonoid group (dementia patients/total people in the group)/risk of dementia in the low flavonoid group (dementia patients/total people in the group)) between the highest and lowest tertiles of flavonoid intake was 0.49.

A very large epidemiological study (total of ~130,000 people followed for 20–22 years) published several years ago [18] examined whether higher intakes of total flavonoids were associated with a lower risk of PD. Five major sources of dietary flavonoids (tea, berry fruits, apples, red wine, and orange or orange juice) were examined using a food composition database and a food frequency questionnaire. In men, those with the highest quintile of flavonoid consumption had a 40% lower risk of developing PD as compared to those in the lowest quintile. However, a significant relationship between overall flavonoid consumption and PD risk was not seen in women.

Flavonoids were historically characterized on the basis of their antioxidant and free radical scavenging effects. However, more recent studies have shown that flavonoids have a wide range of activities that could make them particularly effective for blocking the age-associated toxicity pathways associated with neurodegenerative diseases [19,20,21,22].

In the following paragraphs, the results of pre-clinical, and in a few cases, clinical studies, that looked at the beneficial effects of different flavonoids in animal models of AD, PD, HD, or ALS, and when information is available, their possible modes of action, will be described. Although the goal was to be as comprehensive as possible, some studies may have been missed inadvertently. Several recent reviews also cover a subset of these topics [20,21,22]. There are no studies on flavonoids in models of FTD, although mouse models of the disease do exist [23].

For each disease, a brief overview is given followed by a description of the models used to study the effects of flavonoids on the disease, and then the flavonoid results based on the subclasses of flavonoids are discussed. The focus is primarily on studies employing single flavonoids and the analysis mainly utilizes primary reports.

### 1.5. Flavonoids and Alzheimer’s Disease (AD)

Alzheimer’s disease is the most common type of dementia. It is characterized pathologically by the presence of both extracellular neuritic plaques containing amyloid beta (Aβ) peptide and intracellular neurofibrillary tangles containing tau [24]. Clinically, AD results in a progressive loss of cognitive ability and eventually daily function activities [25,26]. Current approved therapies are only symptomatic, providing moderate improvements in memory without altering the progression of the disease pathology [27,28]. Although a large number of clinical trials have been conducted in recent years with drug candidates designed to directly or indirectly reduce the amyloid plaque load, all of these trials have failed [29].

Three different types of models have been used to study the possible beneficial effects of flavonoids in AD: interventional, transgenic, and sporadic. For the interventional studies, Aβ is injected directly into the cerebral ventricles in the brains of the rodents (intracerebroventricular (icv) injection). There are numerous transgenic models of AD that are based on the mutations associated with the rare genetic form of the disease (familial AD or FAD). The models develop different degrees of cognitive impairment, levels of plaques and tangles, synaptic loss, gliosis, and nerve cell death depending on the type and number of mutations (1–5) [30]. Although AD drug discovery has largely focused on these FAD models, this form of the disease accounts for only ~1% of the total cases [31], and may be quite distinct from the much more prevalent, old-age-associated, sporadic form of AD. Importantly, while many therapies directed against the amyloid pathway are effective in FAD transgenic mice, to date none has translated into the clinic [29]. Since old age is by far the greatest risk factor for AD [31,32], animal models that incorporate aging into disease development may prove more useful for the development of therapies. One mouse model of aging that also develops characteristics of AD is the senescence-accelerated prone 8 (SAMP8) mouse that was developed in Japan by selective breeding of a rapidly aging phenotype. These mice exhibit a progressive, age-associated decline in brain function similar to human AD patients [33,34,35]. As they age, SAMP8 mice develop an early deterioration in learning and memory, as well as a number of pathophysiological alterations in the brain, including increased oxidative stress, inflammation, vascular impairment, gliosis, Aβ accumulation, and tau hyperphosphorylation. Using an integrative multiomics approach, we recently identified a number of behavioral and physiological changes that are altered with aging in these mice [36]. Although much less widely used, the senescence-accelerated OXYS rat also spontaneously develops all of the brain changes associated with AD, including structural alterations, neuronal loss, Aβ accumulation, tau hyperphosphorylation, and cognitive impairment [37].

The flavone 7,8-dihydroxyflavone (7,8-DHF) has been tested by several labs in the 5 x FAD model (multiple AD-linked mutations in the amyloid precursor protein (APP) and presenilin 1 (PS1)). Improvement in performance in the Y maze, a test for working memory, was seen following short term (10 days) intraperitoneal (ip) injection of 7,8-DHF (5 mg/kg) into 12–15 months old 5 x FAD mice [38]. Chronic oral administration of 7,8-DHF (5 mg/kg/day) from 2–6 months of age in this same model also improved memory and reduced synapse loss [39]. In contrast, in a 2 x FAD model (AD-linked mutations in APP and PS1), daily ip administration of 7,8-DHF (5 mg/kg) for 4 weeks beginning at 6 weeks of age showed no effect on learning and memory impairments in the Morris water maze [40]. However, in the first two studies [38,39], clear activation of TrkB, the proposed target of 7,8-DHF, was seen, while in the third study [40], this was not examined. Thus, the lack of effects on memory could be due to a failure to activate TrkB in this study. Further studies on this flavone and AD are clearly warranted.

The flavone apigenin has been tested in a 2 × FAD model (AD-linked mutations in APP and PS1), where oral administration (40 mg/kg/day) for 12 weeks beginning at 4 months of age resulted in improved learning and memory, reduced deposition of insoluble Aβ, a decrease in markers of oxidative stress, and an increase in the activity of the ERK-CREB pathway, an indicator of neurotrophic activity [41].

In a 1 × FAD model (AD-linked mutation in APP), a four month daily ip treatment with the citrus flavone nobiletin (10 mg/kg) improved memory and reduced the levels of both soluble and insoluble Aβ [42]. Consistent with these results, three months of daily ip injections of nobiletin (10 and 30 mg/kg) starting at 6 months of age in the 3 × FAD model (AD-linked mutations in APP, PS1, and tau) resulted in an improvement in memory on multiple tests and a reduction in soluble Aβ levels, as well as reactive oxygen species (ROS) in the mice treated with 30 mg/kg [43]. Similarly, 2 months of daily ip injections of nobiletin (10 and 50 mg/kg) starting at 4–6 months in the SAMP8 mice improved memory in multiple assays and reduced some markers of oxidative stress at both doses [44].

Daily ip injections of the flavone baicalein (10 mg/kg) beginning at 6 months of age also prevented deficits in working memory and reduced the production of Aβ in a 1 x FAD model (AD-linked mutation in APP) [45].

The flavonol fisetin has been tested in all three types of AD models (icv Aβ injection, 2 × FAD mice, SAMP8 mice), where it consistently prevented the loss of cognitive function [46,47,48]. Both oral administration (25 mg/kg/day) [46,47] and daily ip injections (20 mg/kg) [48] proved effective. In all of the models, fisetin was found to maintain the levels of synaptic proteins and to reduce markers of inflammation. It also reduced markers of oxidative stress and particularly lipid peroxidation and activated the ERK pathway, which is involved in both memory [49] and neurotrophic factor production and signaling [50]. However, the effects on the levels of soluble and insoluble Aβ varied between the different models, suggesting that this may not be the critical target.

Another flavonol, quercetin, was also shown to have similar benefits in multiple models of AD [51] following either oral administration (SAMP8) (25 mg/kg/day) [52] or ip injection (25 mg/kg every 2 days) (3 x FAD) [53]. Similar to fisetin, not only was quercetin able to reduce cognitive impairment but also modulated multiple targets in the brains of the treated mice, including the levels of soluble and insoluble Aβ and the activation of astrocytes or microglia, indicators of an on-going inflammatory response. Interestingly, in the study on SAMP8 mice [52], the effects of the administration of free quercetin (25 mg/kg/day) to those of nanoencapsulated quercetin particles (25 mg/kg/every 2 days) were compared. An almost 2-fold higher level of quercetin was found in the brains of the quercetin nanoparticle-treated mice, which correlated with significant effects on learning and memory, as well as astrogliosis [52]. Rutin, a glycoside of quercetin that combines quercetin with rutinose, was also found to have beneficial effects in rats injected icv with Aβ [54]. Daily ip injection of rutin (100 mg/kg) for 3 weeks after Aβ administration prevented memory loss, reduced lipid peroxidation, and increased markers of neurotrophic factor activity [54]. A recent review covered some of these same studies in more detail [20].

Similar to the results with the quercetin nanoparticles, it was found that daily ip injection of a mixture of anthocyanins (glycosylated form of anthocyanidins) from Korean black soybeans encapsulated in gold nanoparticles (10 mg/kg/day) were much more effective at reducing memory impairments, loss of synaptic proteins, and neuroinflammation in icv Aβ-injected mouse brains than the free anthocyanins [55,56].

Non-fermented teas, such as green tea, contain high levels of catechins (flavanols), including (−)-epigallocatechin gallate (EGCG), (−)-epigallocatechin, (−)-epicatechin gallate, (−)-epicatechin, and (+)-catechin. Studies on EGCG in both a 1 x FAD mouse model (50 mg/kg/day) [57] and the SAMP8 mouse (15 mg/kg/day) [58] found that long-term, oral administration improved cognitive function, reduced the levels of soluble Aβ, and prevented the decrease in some synaptic proteins. In addition to these animal studies, over 10 clinical studies have been conducted on green tea and AD [59]. These include cross-sectional and longitudinal studies where the frequency of drinking green tea and cognitive function were assessed either at a single time point or over time and interventional studies where participants were given a green tea extract and followed over time. Most, but not all, of the longitudinal and cross-sectional studies showed an inverse relationship between green tea consumption and cognitive dysfunction. Furthermore, a meta-analysis showed a dose-dependent effect of green tea consumption on cognitive impairment. However, the interventional studies had many fewer participants and the results were less consistent.

Other flavonoids that have shown benefits in animal models of AD include the flavanone glycoside hesperidin (100 mg/kg/day) [60,61] and the isoflavone puerarin (30 mg/kg/day) [62].

In summary, multiple flavonoids have shown significant benefits in three distinct models of AD (Table 1). All of the flavonoids described above improved cognitive function, and where examined, reduced markers of inflammation, oxidative stress, and synaptic dysfunction, and increased neurotrophic factor signaling. In addition, many reduced the accumulation of soluble or insoluble Aβ. Together these results support the idea that multi-target compounds, such as flavonoids that act on several different pathophysiological changes that occur in the aging brain and that are exacerbated in AD, have a strong potential for the treatment of the disease. Unfortunately, very few human studies have been performed, so whether this potential will ever be realized is not clear. In addition, the significantly enhanced effects seen with the nanoparticles of quercetin [52] and anthocyanins [55,56] strongly suggest that if flavonoids are to be used pharmacologically, then the formulation needs to be more carefully considered.

### 1.6. Flavonoids and Parkinson’s Disease (PD)

Parkinson’s disease (PD) is a chronic, progressive neurodegenerative disease and the second most common neurodegenerative disease after Alzheimer’s. The characteristic features of PD include resting tremor, bradykinesia (slowness of movement), rigidity, and postural instability [63]. PD is also associated with a variety of non-motor symptoms that contribute to disability. The majority of PD cases are sporadic with only about 10% of PD patients reporting a family history of the disease [64]. Age is the greatest risk factor for disease development. The pathological hallmark of PD is the degeneration of the dopaminergic (DA) neurons in the substantia nigra pars compacta (SNc) [63]. Since these neurons synapse with neurons in the striatum, their demise leads to the depletion of striatal dopamine. PD is also characterized by the presence of cytoplasmic protein aggregates, called Lewy bodies, in the remaining DA neurons of the SNc. Currently, there is no test to diagnose PD prior to the onset of motor symptoms and available treatments only improve the symptoms but do not stop disease progression. Importantly, by the time that the first symptoms appear, striatal dopamine is reduced by ~80%, and ~60% of the DA neurons of the SNc have died [65]. Thus, both better methods of diagnosis and treatments that can begin before overt disease onset are needed.

Animal models of PD generally involve treatment with a toxin, such as a pesticide or other toxic compound that has been associated with PD in vivo. The two most widely used models are 6-hydroxydopamine (6-OHDA) and 1-methyl-4-phenyl-1,2,3,6-tetrahydropyridine (MPTP) [66,67]. Both the herbicide paraquat and the pesticide rotenone have also been used to model PD. However, none of these models recapitulates all of the aspects of human PD [66,67] and most are very rapid onset, as compared to the age-dependent development of PD in human patients. Although animal models in which one or more of the genes associated with familial PD are mutated have been developed [67], most of these genetic PD models lack nigrostriatal degeneration and there is also a problem with inconsistent phenotypes between different mouse lines with the same mutation [67]. Thus, they have not been used extensively for testing of potential therapeutic compounds.

Quite a large number of different flavonoids from most of the different classes have been tested in the different rodent toxin models of PD (Table 1). Some of these results have been recently reviewed [22] and these and others are described below.

The flavone baicalein has been tested in several different models in rodents, including the MPTP model using both ip injection in rats (40 mg/kg/day) [68] and oral administration in mice (200 mg/kg/day) [69], and the rotenone model using ip injection in rats (2.5 mg/kg/day) [70]. In all cases, baicalein attenuated the loss of DA neurons, while in the mouse MPTP model and the rotenone model, it also reduced behavioral impairments and markers of oxidative stress [70]. In addition, in the MPTP models, it reduced markers of inflammation [68,69].

The flavone 7,8-DHF has also been tested in several different PD models. Oral pre- and post-treatment (12-16 mg/kg/day) in the 6-OHDA model in rats improved behavior and reduced the loss of DA neurons in the SNc [71]. Pre- and post-treatment ip injection of 7,8-DHF (5 mg/kg/day) also reduced motor function impairment and prevented DA neuron loss in the MPTP model in mice [72]. The 7,8-DHF was also able to prevent further decreases in motor function and tyrosine hydroxylase (TH) levels when it was given by ip injection (5 mg/kg/day) after MPTP treatment in a slower model of disease progression in mice [73]. Similarly, oral administration of 7,8-DHF (30 mg/kg/day) prevented the MPP+-induced progressive loss of DA neurons in a monkey model of PD [74]. The 7,8-THF was also reported to activate the TrkB receptor, thereby activating neurotrophic factor signaling pathways, and all of the rodent studies [71,72,73] showed a maintenance of TrkB activation by 7,8-DHF in the presence of the different toxins.

Several other flavones have also shown protective effects in PD models, including apigenin using both ip injection (10 and 20 mg/kg/day) in the rotenone model in rats [75] and oral administration (5, 10, and 20 mg/kg/day) in the MPTP model in mice [76], oral administration of chrysin (10 mg/kg/day) in the 6-OHDA model in mice [77], oral administration of luteolin (10 and 20 mg/kg/day) in the MPTP model in mice [76], oral administration of nobiletin (10 mg/kg/day only) in the MPTP model in rats [78], and daily ip injection of morin (50 mg/kg) in the MPTP model in mice [79]. All five flavones helped to preserve the DA neurons and reduced markers of inflammation, while apigenin, chrysin, and luteolin prevented toxin-mediated decreases in neurotrophic factor gene expression. Apigenin, chrysin, luteolin, and morin also reduced toxin-induced behavioral alterations.

The flavonol quercetin has also been tested in several different models, and except for a study using oral pre-administration (20 mg/kg/day) in the 6-OHDA model [80], has shown positive results. However, in a more recent study in the same model, oral administration of quercetin (50 mg/kg/day) did show a beneficial effect where it reduced the loss of striatal dopamine and the increase in markers of oxidative stress [81]. Using the rat model of rotenone toxicity [82], quercetin was found to attenuate striatal dopamine depletion in a dose-dependent manner when given by ip injection (50 and 75 mg/kg/day) for 4 days after the administration of the toxin. Quercetin also reduced rotenone-induced behavioral changes and the loss of tyrosine hydroxylase (TH) immunoreactivity in both the SN and striatum. TH immunoreactivity is a marker for the integrity of the nigrostriatal pathway. Oral administration of quercetin (100 and 200 mg/kg/day) prior to the administration of the toxin improved motor function in MPTP-treated mice in a dose-dependent manner [83], which correlated with a significant increase in striatal dopamine levels and a significant decrease in a marker of lipid peroxidation. More recently, quercetin was tested in the MitoPark transgenic mouse model of PD [84]. These mice have a conditional disruption of mitochondrial transcription factor A, specifically in DA neurons, and recapitulate several aspects of human PD, including adult onset, slow impairment of motor function, and degeneration of the nigrostriatal pathway [85]. Oral administration of quercetin (25 mg/kg/day) to these mice for 6–8 weeks beginning at 12 weeks of age moderately but significantly reduced behavioral deficits, striatal dopamine loss, and nigrostriatal degeneration. The quercetin glycoside rutin was also tested in the 6-OHDA model in rats, where daily ip injection (10 and 30 mg/kg) was shown to partially reduce motor deficits when treatment was initiated beginning 3 weeks before administration of the toxin [86]. This correlated with a moderate but significant increase in striatal dopamine levels, as well as an increase in brain GSH levels. In contrast, markers of both lipid and protein oxidation were reduced.

Although the flavonol fisetin (20 mg/kg/day) did not show positive effects in the same 6-OHDA study in rats where quercetin also failed to show a beneficial effect [80], a more recent study using MPTP-treated mice found that oral administration of fisetin (10-25 mg/kg/day) prior to treatment with the toxin dose-dependently increased striatal dopamine levels and largely prevented the loss of TH immunoreactivity in the striatum [87]. Oral administration of the flavonol kaempferol (25, 50, and 100 mg/kg/day) had similar, dose-dependent effects in the same model when started prior to treatment with MPTP [88].

Both the flavonol myricetin (2.5 µg/day) and its glycoside myricitrin (60 mg/kg/day) maintained TH-positive neurons in the 6-OHDA model in rodents when administered by ip (myricitrin) or icv (myricetin) injection [22]. Myricitrin also reduced markers of inflammation and improved motor function, while myricetin increased dopamine levels.

Several flavanols have also shown benefits in PD models. Daily ip injection of catechin (10 and 30 mg/kg) in the 6-OHDA model in rats [89], oral administration of EGCG (25 mg/kg/day) in the MPTP model in mice [90], and oral administration of epicatechin (100 mg/kg/day) in the MPTP model in rats [91] all reduced toxin-induced behavioral deficits. For both catechin and EGCG, these functional improvements correlated with a reduction in striatal dopamine loss.

Flavanones have also shown benefits in PD models, including naringenin in the MPTP model in mice [92] and the 6-OHDA model in rats (oral; 50 mg/kg/day) [80] and hesperetin in the 6-OHDA model in rats [93]. Oral administration of naringenin (25, 50, and 100 mg/kg/day) increased dopamine levels and reduced the loss of TH immunoreactivity, while also lowering markers of inflammation and oxidative stress [92]. The glycoside of naringenin, naringin (ip; 80 mg/kg/day), was tested using both pre- and post-treatment in the 6-OHDA model in rats [94]. While pre-treatment protected against the toxin-induced loss of DA neurons and prevented microglial activation, post-treatment had no beneficial effects [94]. Oral administration of hesperetin (50 mg/kg/day) reduced 6-OHDA-induced behavioral deficits and prevented the loss of DA neurons [93]. These effects correlated with a reduction in some, but not all, inflammatory markers and lower levels of indices of oxidative stress, as well as an increase in GSH levels. Similarly, oral administration of the hesperetin glycoside, hesperidin (50 mg/kg/day), reversed behavioral deficits, reduced striatal dopamine loss, and decreased markers of oxidative stress in the brains of 6-OHDA-treated aged mice [95].

The isoflavones genistein and puerarin have also been tested in rodent PD models. Genistein administered by ip injection improved neuronal survival in both the 6-OHDA model in rats (10 mg/kg/day) and the MPTP model in mice (10 mg/kg/day) [22]. Daily ip injection of puerarin (0.12 mg/kg) reduced DA neuronal loss in the MPTP model in mice, which correlated with decreases in markers of oxidative stress and inflammation and increases in markers of neurotrophic factor signaling [62].

In summary, a wide range of flavonoids have shown significant benefits in multiple rodent toxin-based models of PD. Where examined, they reduced markers of inflammation and oxidative stress and increased markers of neurotrophic factor signaling. Together, these effects contributed to the prevention of nerve cell death and the reduction in behavioral deficits. In addition, many prevented increases in α-synuclein, a protein associated with neuronal damage and death in PD. Thus, similar to the situation with AD, the flavonoids appear to have multiple targets in the PD models, further supporting the idea that multi-target compounds are likely to provide the best treatments for this neurodegenerative disease. However, as genetic models of PD become more reproducible, it will be important to test some of the most promising flavonoids in these models as well to provide further evidence for potential clinical efficacy.

### 1.7. Flavonoids and Huntington’s Disease (HD)

Huntington’s disease is a late onset, progressive, and fatal neurodegenerative disorder characterized by movement and psychiatric disturbances, as well as cognitive impairment. There is, at present, no cure. HD is an autosomally dominant inherited disease that is caused by an unstable expansion of a trinucleotide repeat (CAG) that encodes an abnormally long polyglutamine tract in the huntingtin protein. The age at disease onset inversely correlates with the CAG repeat number. The identification of the disease-causing mutation has allowed the development of a number of cellular and animal models of HD, and these have been used to try to elucidate the mechanisms underlying disease development and progression [96,97,98,99].

Both chemical and genetic rodent models of HD have been used to test the potential preventive role of flavonoids in HD development or progression, although no single model broadly replicates both the behavioral and neuropathological changes seen in humans [100]. The chemical approach uses 3-nitropropionic acid (3-NP), which when administered systemically at low doses to rats or mice causes selective degeneration of striatal neurons—the same neurons that are lost in HD [101]. The genetic models can be divided into three groups: N-terminal transgenic animals that carry only the 5’ portion of the human *HTT* gene, which contains the CAG repeats; full length transgenic animals that carry the full length *HTT* sequence, including the CAG repeats; and knock-in models, in which CAG repeats are engineered directly into the mouse *Htt* genomic locus [100]. These genetic models differ in the time of disease onset and disease severity, with the N-terminal transgenic animals showing the most severe phenotypes.

While there have only been a limited number of studies on the potential beneficial effects of flavonoids in HD models, the published results suggest that specific flavonoids could be of potential clinical use (Table 1). Oral administration of the flavone chrysin (50 mg/kg/day) improved behavior and reduced markers of oxidative stress and cell death, and enhanced the survival of striatal neurons in the 3-NP model of HD in rats [102].

Chronic oral administration of 7,8-DHF (5 mg/kg/day) to the R6/1 N-terminal transgenic mouse model of HD delayed the development of motor and cognitive deficits, prevented the loss of striatal volume, enhanced a marker of neurotrophic factor signaling, and reduced some markers of inflammation [103]. However, the effects on lifespan, which is greatly reduced in this mouse model of HD, were not assessed.

The flavonol fisetin was tested in the R6/2 mouse model of HD, which like the R6/1 model, is a N-terminal transgenic line that has an aggressive disease phenotype and shortened lifespan [104]. Fisetin was fed to genotyped R6/2 mice and their wild type littermates in the food beginning at ~6 weeks of age (25 mg/kg/day). The mice were tested on the rotarod from ~7–13 weeks of age and survival was followed. At the time of acquisition of the animals, rotarod performance was already impaired in the R6/2 mice as compared to their wild type littermates. Motor performance in the rotarod test declined significantly more rapidly in the animals on the control diet as compared to those on the fisetin diet. Similarly, while the median life span of the R6/2 mice on the control diet was 104 days, that of fisetin-fed mice was increased by ~30% to 139 days. The in vivo mechanisms underlying the effects of fisetin were not explored in this study.

The closely related flavonol quercetin was tested by two different groups in the 3-NP model in rats. In the first study, which used oral administration [105], quercetin (25 mg/kg/day) was found to reduce motor deficits, improve mitochondrial function, and attenuate some markers of oxidative stress. Although there was some suggestion of beneficial effects on striatal neuronal survival, the results were not clear. In the second study, which used daily ip injection of quercetin (50 mg/kg) [106], it was also found to improve motor function, as well as reduce a marker of inflammation, but it did not prevent the loss of striatal neurons. The quercetin glycoside, rutin, was also tested in the 3-NP model in rats but using a different protocol for 3-NP treatment in conjunction with oral administration [107]. Similar to quercetin, rutin (25 and 50 mg/kg/day) prevented 3-NP-induced impairments in motor function and decreased markers of inflammation. It also reduced markers of oxidative stress.

In contrast to quercetin, daily ip injection of the flavonol kaempferol (25 mg/kg) in 3-NP-treated rats not only reduced motor deficits but also attenuated the loss of striatal neurons [108]. These effects correlated with a reduction in markers of oxidative stress.

The flavanone glycoside hesperidin was also tested in the 3-NP model in rats [109] using oral administration (100 mg/kg/day). Similar to the other flavonoids in this model, hesperidin reduced motor deficits, as well as markers of inflammation and oxidative stress. The isoflavone genistein was also found to be protective in the 3-NP model in rats when given by daily ip injection (10 and 20 mg/kg) [110]. Genistein also reduced motor deficits and decreased markers of oxidative stress, inflammation, and nerve cell death.

Dietary supplementation with mixed berry anthocyanins (~100 mg/kg/day) was shown to delay the loss of motor function in the R6/1 N-terminal transgenic mouse model of HD [111]. However, this effect was only seen in female HD mice. The effects on lifespan were not examined.

In summary, a number of different flavonoids have shown benefits, particularly with regard to preserving motor function, in both chemical and transgenic models of HD. However, since most, if not all, of the studies with the 3-NP model involve pretreatment with the flavonoid, it is not clear if some of the effects of the flavonoids could be due to directly inhibiting the actions of 3-NP itself rather than reducing the consequences of 3-NP treatment. Thus, it would be worth testing those flavonoids that showed promise in the 3-NP assay in a transgenic model of HD. Similar to the effects of the flavonoids in AD models, the flavonoids appear to have multiple targets in the HD models, including reducing markers of inflammation and oxidative stress.

### 1.8. Flavonoids and Amyotrophic Lateral Sclerosis (ALS)

ALS is a fatal neurodegenerative disease that is characterized by the loss of the motor neurons that control the voluntary movement of muscles, resulting in paralysis and death, usually within 5 years of a diagnosis [112]. Approximately 10% of ALS cases are due to heritable gene mutations but different gene mutations are increasingly being found in patients with no family history of ALS, suggesting that the genetic component is more complicated than originally thought [113]. Moreover, there are overlaps between ALS and FTD [114]. Although there are three FDA-approved drugs for ALS, they all have very modest effects on survival (https://alsnewstoday.com/approved-treatments/) (access on 21 June 2019).

The most commonly used mouse model of ALS is based on Cu/Zn superoxide dismutase 1 (*SOD1*), the first gene mutation that was shown to cause ALS [113]. SOD1 mutations are found in ~20% of patients with the familial form of ALS. The most extensively used form of these mouse models is in the SOD1-G93A transgenic mouse. Although the different SOD1 transgenic mouse lines are not identical, they all show protein aggregation, motor neuron loss, axonal denervation, progressive paralysis, and reduced lifespan [113].

Despite the absence of effective treatments for ALS and the promising results with flavonoids in other neurodegenerative diseases as described above, very few flavonoids have been tested in animal models of ALS (Table 1). In all studies, the SOD1-G93A model was used.

Three times per week ip injection of the SOD1-G93A mice with 7,8-DHF (5 mg/kg) beginning at 1 month of age reduced the age-dependent decrease in motor performance and preserved total motor neuron count and dendritic spine density on motor neurons [115]. However, effects on lifespan were not examined.

Oral administration of the flavonol fisetin (9 mg/kg) beginning at 2 months of age significantly delayed the development of motor deficits, reduced their rate of progression, and increased lifespan [116]. This correlated with a significant increase in the motor neuron count in the spinal cord. At the molecular level, fisetin increased the levels of both phospho-ERK and the antioxidant protein heme oxygenase 1. Interestingly, fisetin also increased ERK phosphorylation in a transgenic AD model [46] and in HD flies [104], suggesting that this may at least partly contribute to its beneficial effects in these different models of neurodegenerative diseases.

Several studies have shown that oral administration of the flavanol EGCG (5.8–10 mg/kg) [117,118] can also delay symptom onset and extend lifespan in the SOD1-G93A mice. Consistent with these observations, EGCG increased motor neuron survival. These effects were correlated with a decrease in multiple markers of inflammation.

In summary, while there have been few studies with flavonoids in models of ALS, the published results suggest that this is an area that warrants further exploration, especially as all of the flavonoids that have shown benefits in the transgenic ALS model have also had positive effects in other age-associated neurodegenerative diseases.

## 2. Summary and Outlook

A number of different flavonoids from all of the six groups have been shown to have beneficial effects in models of AD, PD, HD, and ALS (Table 1). While many flavonoids have only been tested in models of one neurodegenerative disease, others, such as fisetin and 7,8-DHF, have shown efficacy in models of all four of the diseases highlighted in this review. These results strongly support the idea that common changes associated with the aging brain underlie the development of these diseases and that compounds that can address these changes have the best chance of clinical success. These changes include increases in oxidative stress, alterations in protein processing, decreases in neurotrophic factor signaling, synaptic dysfunction, increased inflammation, and cell death, which together contribute to behavioral impairments and cognitive dysfunction (Figure 2). As discussed in this review, flavonoids have the potential to reduce or prevent all of these changes. However, it appears that more work is needed before these compounds are taken seriously as possible therapeutics for neurodegenerative disease treatment. This includes developing better approaches to administration, such as nanoparticles [52,55,56] or other types of formulations [119] that will improve their ability to get into the brain and comparing different flavonoids head to head in the same model in order to determine which ones might have the best chance of clinical success. In addition, there may be synergism between the actions of some of the flavonoids, and this possibility is worth exploring both in vitro and in vivo.

## Figures and Tables

**Figure 1 ijms-20-03056-f001:**
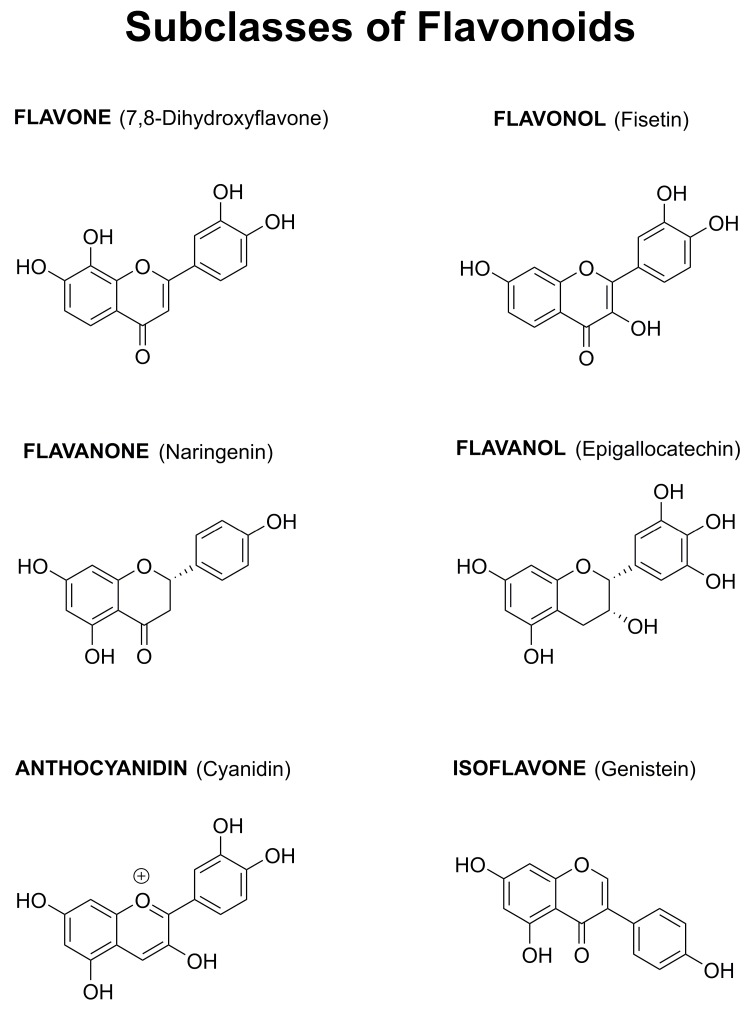
Structures of Representative Flavonoids from the Six Classes.

**Figure 2 ijms-20-03056-f002:**
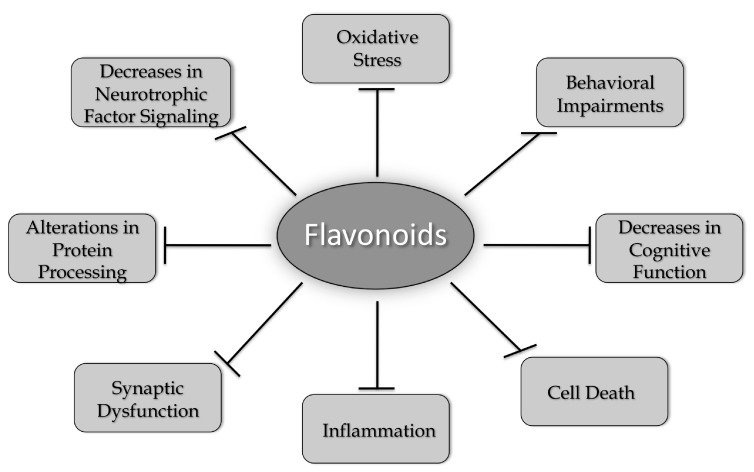
Flavonoids alter multiple pathways implicated in brain aging and neurodegenerative diseases. As discussed in this review, flavonoids can increase brain cell function and neuronal survival by reducing oxidative stress, activating neurotrophic factor signaling pathways, preventing alterations in protein processing, reducing synaptic dysfunction, and inhibiting inflammatory responses. Flavonoids can also enhance cognitive function and modulate behavioral impairments. Therefore, they have the potential to act as multi-factorial therapeutics for reducing the impact of neurodegenerative diseases.

**Table 1 ijms-20-03056-t001:** Flavonoids that have shown efficacy in preclinical models of Alzheimer’s disease (AD), Parkinson’s disease (PD), Huntington’s disease (HD), or amyotrophic lateral sclerosis (ALS).

	AD	PD	HD	ALS
**Flavones**				
7,8-DHF	X	X	X	X
Apigenin	X	X		
Baicalein	X	X		
Chrysin		X	X	
Luteolin		X		
Morin		X		
Nobiletin	X	X		
**Flavonols**				
Fisetin	X	X	X	X
Kaempferol			X	
Myricetin		X		
Myricitrin		X		
Quercetin	X	X	X	
Rutin	X	X	X	
**Flavanols**				
Catechin		X		
Epicatechin		X		
ECGC	X	X		X
**Flavanones**				
Hesperetin		X		
Hesperidin	X	X	X	
Naringenin		X		
Naringin		X		
**Anthocyanidins**				
Anthocyanins	X		X	
**Isoflavones**				
Genistein		X	X	
Puerarin	X	X

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
