# Peer review of "The Potential of Flavonoids for the Treatment of Neurodegenerative Diseases"

_ijms, 2019, doi:10.3390/ijms20123056_

Round 1
Reviewer 1 Report
In this review article, authors present a review on the potential benefits of flavonoids on four neurodegenerative diseases: Alzheimer’s disease, Parkinson’s disease, Huntington’s disease and amyotrophic lateral sclerosis. Four diseases combined affect more than 6 million people in the USA, and there is no effective treatment currently available for any of them. After reviewing published literature, the author found that a number of flavonoids have been shown to have some beneficial effects in animal models of the four neurodegenerative diseases.
The review is sound and well written. Unfortunately, although the use of flavonoids against neurodegenerative diseases has shown to have some beneficial effects, overall they have not gone beyond animal testing due to insufficient health improvement observed. This review article could be published with minor modifications in a MDPI journal of lower impact factor than IJMS.
Several minor issues should be addressed.
1. The abstract mentions the word “disease” in three consecutive lines: 12, 13 and 14. The author could find a way to spread the use of the same word.
2. Punctuation signs would need a review. For instance, in line 40 there should be a coma after the abbreviation (ALS). The coma after “if not” should be moved after “more” instead.
3. Keywords should include the term “neurodegenerative disease”
4. At the end of line 31, “very nice” could be replaced by “comprehensive.”
5. Section 2.4, line 126. Authors should explain the meaning of “risk ratio,” and how it is calculated.
6. Line 142. Author should try to avoid the use of first person of singular “I.”
7. There should be a dash between numbers and chemical names. For instance, 7,8-dihydroxyflavone (or 7,8-DHF). That must be corrected throughout the article.
8. The caption of Table 1 should be moved to the top of the table and not be repeated.
9. Table 1 should be improved. Instead of only a check mark, table 1 should include the type of study, main results and corresponding references.
10. The text (not only the abbreviation section at the end) should also explain the meaning of the term MPTP.
11. Line 303. The abbreviation TH is used before explaining the meaning (explanation can be found in line 324).
12. Line 469 should use the abbreviation 7,8-DHF instead of full name for 7,8-dihydroxyflavone.
Author Response
Reviewer #1:
Thank you for the helpful comments. Below are my responses.
1. The abstract mentions the word “disease” in three consecutive lines: 12, 13 and 14. The author could find a way to spread the use of the same word.
I have used a few synonyms for disease in the Abstract as requested.
2. Punctuation signs would need a review. For instance, in line 40 there should be a coma after the abbreviation (ALS). The coma after “if not” should be moved after “more” instead.
I have changed the punctuation as requested.
3. Keywords should include the term “neurodegenerative disease”
I have included this term as a keyword.
4. At the end of line 31, “very nice” could be replaced by “comprehensive.”
I have made this change.
5. Section 2.4, line 126. Authors should explain the meaning of “risk ratio,” and how it is calculated.
I have included an explanation of risk ratio and how it is calculated.
6. Line 142. Author should try to avoid the use of first person of singular “I.”
This section has been rephrased.
7. There should be a dash between numbers and chemical names. For instance, 7,8-dihydroxyflavone (or 7,8-DHF). That must be corrected throughout the article.
This change was made throughout the manuscript.
8. The caption of Table 1 should be moved to the top of the table and not be repeated.
This has been corrected.
9. Table 1 should be improved. Instead of only a check mark, table 1 should include the type of study, main results and corresponding references.
The purpose of Table 1 is to provide an overview of the flavonoids that have been tested in the different preclinical disease models. Including more details would defeat this purpose. I have included the details of all of the studies in the text and have added the doses of the flavonoids used in the studies to the text as requested by Reviewer #3.
10. The text (not only the abbreviation section at the end) should also explain the meaning of the term MPTP.
This had been corrected.
11. Line 303. The abbreviation TH is used before explaining the meaning (explanation can be found in line 324).
This has been corrected.
12. Line 469 should use the abbreviation 7,8-DHF instead of full name for 7,8-dihydroxyflavone.
This has been corrected.
Reviewer 2 Report
this review put together main information on neuroiprotective effects of flavonoids in four, most common neurodegeneative diseases. Most of the reviews concentrate on more detailed mechanisms of these disease but in this case, where the discussed substances have sucha broad spectrum of action, it is not possible to concentrate on one element. The author presents information concerning different aspects of neiroprotective effect of flavonoids, from oxidative stress, through alternations in many signaling pathways and direct death actions. All this is supported by the recent references on the subject. The construction of the manuscript is simple, making it easy to read and the summary section is supported by the information presented in the manuscript.
Author Response
Thank you for the nice comments.
Reviewer 3 Report
In this review, the author mentions the evidence for beneficial effects of multiple flavonoids in models of AD, PD, HD and ALS, and common mechanisms of flavonoids. These preclinical data strongly supports further investigation of specific flavonoids for the treatment of neurodegenerative diseases. In addition, the author did a well review of neurodegenerative diseases and the efficacy of flavonoids in these diseases in table I. However, there is no dosage of flavonoids mentioned in this article, since the concentrations of flavonoids in animal models are very important information for clinical research, please add some information about dosage of flavonoids in preclinical models. Some other areas that need to be improved in this manuscript are outlined below.
1. In table I, please add dosage of flavonoids and references.
2. In table I, is there any effect of Epicatechin on these 4 models?
Author Response
Reviewer #3:
Thank you for the helpful comments. My responses are listed below.
please add some information about dosage of flavonoids in preclinical models.
I have now included all of the dosage information in the text.
1. In table I, please add dosage of flavonoids and references.
As noted in response #9 to Reviewer #1, the purpose of Table 1 was to provide an overview of the results with the flavonoids. The dosage and references are all in the text. Including that information in the table would defeat the purpose which was to allow readers to quickly see which flavonoids have been tested in the different models of neurodegenerative diseases.
2. In table I, is there any effect of Epicatechin on these 4 models?
Yes, this omission has been corrected.
Round 2
Reviewer 1 Report
As stated in my previous review, the manuscript is sound and well written. Additionally, it has been edited with the suggested minor corrections. However, results of flavonoids against neurodegenerative disorders do not seem to be relevant enough to merit publication in a 3.7 impact factor journal. I would suggest the authors to publish in a lower impact factor journal.